# Pseudo Labels for Unsupervised Domain Adaptation: A Review

**Yundong Li \*** , **Longxia Guo and Yizheng Ge**

School of Information Science and Technology, North China University of Technology, Beijing 100144, China; lorraine@mail.ncut.edu.cn (L.G.); gyz@mail.ncut.edu.cn (Y.G.)
\* Correspondence: liyundong@ncut.edu.cn; Tel.: +86-010-88803014

**Abstract:** Conventional machine learning relies on two presumptions: (1) the training and testing datasets follow the same independent distribution, and (2) an adequate quantity of samples is essential for achieving optimal model performance during training. Nevertheless, meeting these two assumptions can be challenging in real-world scenarios. Domain adaptation (DA) is a subfield of transfer learning that focuses on reducing the distribution difference between the source domain ($\mathcal{D}_s$) and target domain ($\mathcal{D}_t$) and subsequently applying the knowledge gained from the $\mathcal{D}_s$ task to the $\mathcal{D}_t$ task. The majority of current DA methods aim to achieve domain invariance by aligning the marginal probability distributions of the $\mathcal{D}_s$. and $\mathcal{D}_t$. Recent studies have pointed out that aligning marginal probability distributions alone is not sufficient and that alignment of conditional probability distributions is equally important for knowledge migration. Nonetheless, unsupervised DA presents a more significant difficulty in aligning the conditional probability distributions because of the unavailability of labels for the $\mathcal{D}_t$. In response to this issue, there have been several proposed methods by researchers, including pseudo-labeling, which offer novel solutions to tackle the problem. In this paper, we systematically analyze various pseudo-labeling algorithms and their applications in unsupervised DA. First , we summarize the pseudo-label generation methods based on the single and multiple classifiers and actions taken to deal with the problem of imbalanced samples. Second, we investigate the application of pseudo-labeling in category feature alignment and improving feature discrimination. Finally, we point out the challenges and trends of pseudo-labeling algorithms. As far as we know, this article is the initial review of pseudo-labeling techniques for unsupervised DA.

**Keywords:** pseudo-labeling; unsupervised domain adaptation; feature alignment; deep learning; transfer learning

## 1. Introduction

Deep learning has achieved remarkable success in diverse fields, including object detection, speech recognition, health care, and computer vision in bygone years. Its effectiveness is heavily dependent on a substantial quantity of training data, yet collecting vast labeled data is challenging, costly, and time intensive. Meanwhile, the model's performance can be compromised when dealing with new domains owing to domain shifts. Hence, it is a significant and arduous task to maximize the utilization of existing labeled data to boost the model's generalization capability and compensate for the sample scarcity.

To address the above issues, the research field of domain adaptation (DA) was established. DA endeavors to migrate the knowledge acquired from labeled data in the source domain ($\mathcal{D}_s$) to the target domain ($\mathcal{D}_t$), with the purpose of enhancing the model's performance in the $\mathcal{D}_t$ [1–5]. Tan et al. [1] provided a definition of deep transfer learning and divided it into four groups: adversarial, instance-based, network, and mapping. Mei et al. [2] present a comprehensive survey of deep DA methods and categorize them according to their loss functions. Wilson et al. [3] discuss various DA methods in single-source DA. Kouw et al. [4] divide DA into three categories in terms of how classifiers learn from the $\mathcal{D}_s$ and generalize to the $\mathcal{D}_t$: methods based on a single observation (sample-based), methods based on the set of observations representation (feature-based) methods, and parameter

estimator-based (inference-based) methods. Fan et al. [5] classified DA into different types based on the label sets in the $\mathcal{D}_t$ and $\mathcal{D}_t$, including open-set, close-set, partial, generalized, and zero-shot DA.

The target of DA is to boost a model that can generalize well to the $\mathcal{D}_t$ that are related but not identical to the $\mathcal{D}_s$ by leveraging the knowledge learned from the $\mathcal{D}_s$ [6]. The main challenge in DA is addressing the distribution shift stemming from the disparities in feature distributions between the $\mathcal{D}_s$ and $\mathcal{D}_t$. DA can be classified as supervised DA, semi-supervised DA, and unsupervised DA, derived from whether the $\mathcal{D}_t$ is labeled or not [2], among which unsupervised DA is the most challenging and is currently receiving a lot of attention from researchers. According to the theoretical analysis of DA by Ben-david et al. [7], the generalization error on the $\mathcal{D}_t$ is defined by (1) the empirical error of the $\mathcal{D}_s$ classifier, (2) the empirical error between the $\mathcal{D}_s$ and $\mathcal{D}_t$, and (3) the ideal joint error. At first, researchers mainly focused on reducing the empirical error between the $\mathcal{D}_s$ and $\mathcal{D}_t$, assuming that the ideal joint error is small. From a probability statistics perspective, this involves aligning the marginal probability distribution between the $\mathcal{D}_s$ and $\mathcal{D}_t$. Such methods include maximum mean difference (MMD) [8], deep adversarial neural networks (DANN) [9], and adversarial discriminative domain adaptation (ADDA) [10]. As the research progresses, more and more researchers have started to focus on the effect of the ideal joint error. Zhang et al. [11] argue that merely aligning the edge distribution between two domains is not enough, and alignment without considering the class-conditional distribution will lead to an increase in the ideal joint error, constraining the upper bound of the model's theoretical error. The majority of existing DA methods only consider aligning the global features of the $\mathcal{D}_s$ and $\mathcal{D}_t$ but fail to ensure the alignment of class-specific features, which may constrain the model's performance on particular tasks. Worse still, inter-domain category-level alignment often requires labels of both domains to achieve, which is difficult for unsupervised DA. Recently, motivated by the pseudo-labeling techniques in semi-supervised learning, an increasing number of researchers have made significant improvements in model performance by assigning pseudo-labels to $\mathcal{D}_t$ to aid in achieving inter-domain category-level alignment. Yet, almost no one has undertaken a systematic organization and analysis of these works, which is the motivation for our review.

We investigated the papers in some top sessions from 2017 to 2022 and counted and analyzed the number of DA papers on pseudo-labeling methods, as shown in Figure 1. Overall, the number of DA-related papers grows year by year, and it is expected that more and more research on DA will be conducted in the coming years. We use darker colors to represent papers on DA that utilize pseudo-labeling methods, and it is evident that the quantity of these papers has been growing steadily over the years. It is worth noting that its share in the overall DA papers is also increasing year by year; in particular, nearly 50% of the DA papers in the ICCV and CVPR conferences in 2022 used pseudo-labeling methods, which indicates the successful application of pseudo-labels in DA.

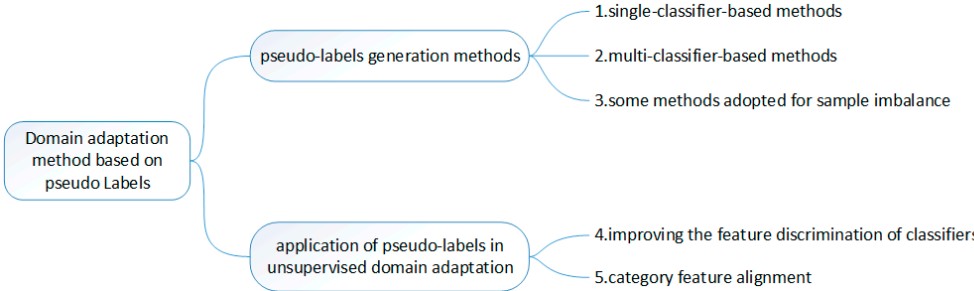

**Figure 1.** Relevant literature statistics of top conferences.

The primary emphasis of this review is to analyze the methods for generating pseudo-labels and their applications in unsupervised DA. First, the generation methods are categorized as either single-classifier-based or multi-classifier-based, and the measures for dealing with sample imbalance are explored. Second, the paper examines the applications of

pseudo-labels in unsupervised DA, which are divided into two parts: using pseudo-labels for category feature alignment and for enhancing the feature discrimination of classifiers.

The structural framework of this paper is shown in Figure 2.

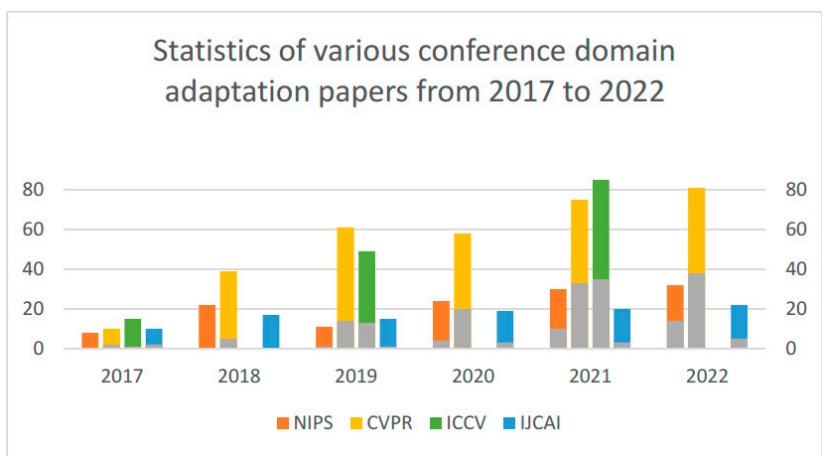

**Figure 2.** The taxonomy of unsupervised DA based on pseudo-labeling.

As far as we know, there are no existing review papers on DA methods based on pseudo-labeling. Specifically, this paper's primary contributions are:

1. We review in detail the background knowledge related to DA and pseudo-labeling methods and sort out the connections and differences between them.
2. We have organized and analyzed the paper in detail in terms of both the pseudo-labeling generation method and the application of pseudo-labeling in unsupervised DA. To the best of our knowledge, it is the first attempt to summarize pseudo labels used in the community of domain adaptation.
3. We conducted a comprehensive review of various pseudo-labeling methods within each category through experimental evaluations. This analysis enables readers to grasp the nuances of each technique and make informed decisions.
4. We point out possible challenges and future directions for pseudo-labeling methods in DA applications.

## 2. Background

### 2.1. Unsupervised Domain Adaptation

In this section, we give a formal definition of unsupervised DA. We denote the $\mathcal{D}_s$ input data and labels as $x^s = \left\{ x_i \right\}_i^{n_s}$ and $y^s = \left\{ y_i \right\}_i^{n_s}$, and the $\mathcal{D}_t$ input data as $x^t = \left\{ x_i \right\}_i^{n_t}$, where $n_s$ and $n_t$ represent the number of samples in the $\mathcal{D}_s$ and $\mathcal{D}_t$, respectively, so that the sample spaces of the $\mathcal{D}_s$ and $\mathcal{D}_t$ can be denoted as $\mathcal{D}_s = \left\{ \left( x_i^s, y_i^s \right) \right\}_i^{n_s}$ and $\mathcal{D}_t = \left\{ x_i^t \right\}_i^{n_t}$, respectively. The feature space and label space of both the $\mathcal{D}_s$ and $\mathcal{D}_t$ are assumed to be identical, $\mathcal{X}_s = \mathcal{X}_t$, $\mathcal{Y}_s = \mathcal{Y}_t$, while the joint probability distributions are different, $P_s(x,y) \neq P_t(x,y)$. The objective of unsupervised DA is to learn a mapping function using the aforementioned data, $f : x^t \mapsto y^t$, to make predictions on the labels in $y^t \in \mathcal{Y}_t$ for the $\mathcal{D}_t$. Interventionary studies involving animals or humans, and other studies that require ethical approval, must list the authority that provided approval and the corresponding ethical approval code.

### 2.2. Pseudo-Labeling

Lee et al. [12] first proposed the method of pseudo-labeling, which uses the token with the highest prediction probability as a pseudo-label, $\hat{y} = \underset{x}{\operatorname{argmax}} f_\theta(x)$, for unlabeled data, and then assigns a weight, $w$, to the unlabeled data and slowly increases it during the training process to perform the training. In contrast to the consistent regularization approach, the pseudo-labeling approach does not rely on region-specific data enhancement and is easier to implement [13]. We categorized the current pseudo-labeling methods

into two types: divergence-based methods and self-training methods. Divergence-based methods utilize multiple networks to perform a task and leverage the divergence of different networks, $f_{\theta_1}(\cdot)$ and $f_{\theta_2}(\cdot)$, to boost the quality of pseudo-labels, thereby improving the overall model's performance [14]. Self-training methods, on the other hand, use the model's own confident predictions to predict the pseudo-label for $\mathcal{D}_t$ unlabeled data, thereby augmenting the training data [13].

## 3. Pseudo-Labeling Generation Methods

Pseudo-labeling generation methods are methods dedicated to improving the accuracy of model-generated $\mathcal{D}_t$ pseudo-labels and to further facilitating domain alignment. We classify the pseudo-labeling generation methods into three categories: single-classifier-based generation methods, multi-classifier-based generation methods, and category-balancing methods for difficult samples. Single-classifier-based generation methods refer to obtaining pseudo-labels of the $\mathcal{D}_t$ by one classifier and completing the DA task by self-training. The multi-classifier-based generation method refers to obtaining more accurate pseudo-labels by the difference of more than two classifiers and then completing the DA task by self-training. The category-balancing method for difficult samples refers to further considering the pseudo-labels category-balancing problem based on the quality of generated pseudo-labels, as a way to obtain higher-quality pseudo-labels.

### 3.1. Single-Classifier-Based Generation Method

The basic assumption of the single-classifier-based generation approach is that the model's own highly confident predictions are correct [15]. The single-classifier-based approach generates pseudo-labels by using the model's own confident predictions for unlabeled data. In semi-supervised classification tasks, it can predict unlabeled data by using a limited quantity of available labeled data, filtering according to some criterion, and finally training the model together with true labels and pseudo-labels [12,16]. In contrast, in unsupervised DA problems, using a model trained from labeled $\mathcal{D}_s$ data and pseudo-labeling the unlabeled $\mathcal{D}_t$ data can be of great help in promoting domain alignment, especially category-level alignment.

In Wang et al. [17], a structured prediction-based selective pseudo-labeling approach was proposed. This method utilizes the structural information of the $\mathcal{D}_t$ data through clustering and labels the $\mathcal{D}_t$ samples collectively based on the clusters they belong to. The distance from the $\mathcal{D}_t$ sample to the cluster center is used as the criterion for pseudo-labeling, with samples closer to the center being more prone to be chosen for pseudo-labeling and for participating in the next round of iterative training. Deng et al. [18] employed a teacher–student-model structure with pseudo-labeling provided by the teacher model. The discriminative learning and category-level alignment goals are achieved through discriminative clustering loss and clustering-based alignment loss.

To address the issue of sparse pseudo-labels generated by single-classifier models, Shin et al. [19] proposed a two-phase pseudo-label densification framework that uses a bootstrapping mechanism in the self-training loss function to boost the model's generalization ability. Wang et al. [20] introduced a binary soft-constrained information entropy to improve the credibility of the mined class prototypes and class anchors, particularly for samples at the decision boundary. This method increased the accuracy of the model in estimating pseudo-labels for the $\mathcal{D}_t$. Zhang et al. [21] proposed AuxSelf-Training for the auxiliary model from the perspective of training samples, in which the sample selection is founded on reducing the proportion of $\mathcal{D}_s$ data and increasing the proportion of the $\mathcal{D}_t$ data proportion to construct the intermediate domain and gradually overcome the distance bias across the domain.

### 3.2. Multi-Classifier-Based Generation Methods

Multi-classifier-based generation methods are extensively used in semi-supervised learning, including the classical approaches proposed by Qin et al. [22] and Zhou et al. [23].

Unlike the single-classifier-based approach, the multi-classifier-based approach usually trains two or three different networks and uses the divergence between different networks to allocate high-quality pseudo-labels to unlabeled samples. These pseudo-labeled samples are then used in training together with the labeled samples, leading to effective DA results.

Inspired by [22,23], Saito et al. [24] proposed asymmetric tri-training for unsupervised domain adaptation (ATDA) for unsupervised DA, as shown in Figure 3.

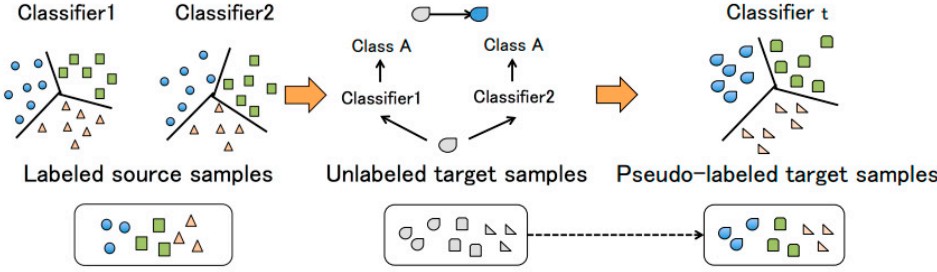

**Figure 3.** The ATDA architecture. (Image: courtesy of Saito et al. [24]).

The approach involves utilizing a shared feature extractor and three classifiers. To enable and to classify from different perspectives, ATDA adds as a regularization term to the cross-entropy loss, as shown in the following Equation (1).

$$E\left(\theta_F, \theta_{F_1}, \theta_{F_2}\right) = \frac{1}{n} \sum_{i=1}^{n} \left[L_y(F_1 \circ F(x_i), y_i) + L_y(F_2 \circ F(x_i), y_i)\right] + \lambda \left|W_1^T W_2\right| \quad (1)$$

where $L_y$ is the cross-entropy loss, and $\theta_{F_1}$ and $\theta_{F_2}$ are hyper-parameters. ATDA screens out the $\mathcal{D}_t$ samples with the consistent sum output of $F_1$ and $F_2$ confidence greater than a certain value to be pseudo-labeled for training. Inspired by ATDA to allocate pseudo-labels to unlabeled $\mathcal{D}_t$ samples and Mixup [25], Li et al. [26] put forward a three-branch CNN model based on an electrocardiogram (ECG), which demonstrated superior performance on the task of classifying heartbeats in the presence of domain shift. Similar to ATDA, Venkat et al. [27] proposed a multi-source DA method that employs pseudo-labels generated by multiple classifiers ground on the consistency of their predictions. This approach achieved promising results in their experiments.

To further mitigate the undesirable consequences of incorrect pseudo-labeling on training, Zheng et al. [28] used uncertainty to mitigate the undesirable consequences of incorrect pseudo-labeling on DA. Unlike the fixed threshold used by Saito et al. [24] and Zou et al. [29], Zheng et al. [28] used a dynamic thresholding approach, as shown in Figure 4.

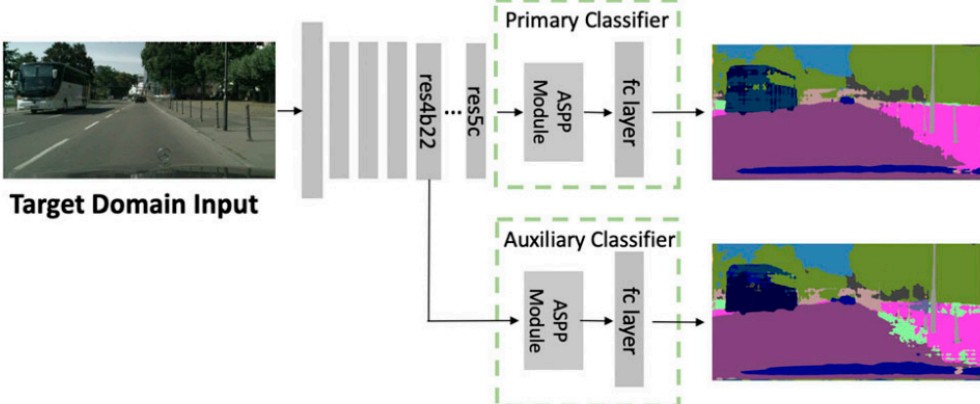

**Figure 4.** The rectifying pseudo-label learning via uncertainty estimation for DA semantic segmentation architecture. (Image: courtesy of Zheng et al. [28]).

The method models the uncertainty of pseudo-labeling by accessing networks with different depths of primary and secondary classifiers to obtain different perspectives and prediction variance. The prediction variance and the classification loss on pseudo-labeling are defined as follows in Equations (2) and (3).

$$D_{kl} = E\left[F\left(x_t^j \big| \theta_t\right) log\left(\frac{F\left(x_t^j \big| \theta_t\right)}{F_{aux}\left(x_t^j \big| \theta_t\right)}\right)\right] \tag{2}$$

$$L_{ce} = E[-\hat{p}_t^j log F(x_t^j \big| \theta_t)] \tag{3}$$

where $F$ is the main classifier and $F_{aux}$ is the secondary classifier. The modified pseudo-label loss function is expressed by Equation (4).

$$L_{rect} = E[exp\{-D_{kl}\}L_{ce} + D_{kl}] \tag{4}$$

When the prediction results of the main classifier and the subclassifier are very different, the value of $D_{kl}$ will be larger, indicating that the pseudo-labels may be inaccurate. Du et al. [30] enhanced the performance of the dual-classifier adversarial training network construction put forward by Saito et al. [31] by introducing additional losses. Specifically, they added a self-supervised loss on the $\mathcal{D}_t$ and a gradient difference loss on both domains on top of the classification loss on the $\mathcal{D}_s$. The self-supervised loss on the $\mathcal{D}_t$ improved the discriminability of the $\mathcal{D}_t$ distribution, which is beneficial for subsequent category-level alignment. In terms of pseudo-label generation, Du et al. [30] used the softmax outputs of two classifiers to weight the samples to obtain the k class prime $c_k$ and finally pseudo-labeled the $\mathcal{D}_t$ data with the nearest prime strategy. Li et al. [32] put forward a method for obtaining accurate pseudo-labels for $\mathcal{D}_t$ data using the prediction consistency of multiple classifiers. The method explicitly adapts the multi-order classifier from the $\mathcal{D}_s$ to the $\mathcal{D}_t$, ensuring that the pseudo-labels are both accurate and diverse. Ge et al. [33] proposed a simultaneous training symmetric network to achieve mutual supervision under collaborative training, thus avoiding the formation of overfitting to the network's own output error, which leads to the amplification of pseudo-labeling noise. Qin et al. [22] added a pseudo-labels training set to the training model of MCD [31] to participate in the training, enhancing the efficacy of this network.

### 3.3. Category-Balancing Methods for Difficult Samples

In the DA problem, the complexity of various DA tasks varies because of domain differences. Similarly, within the same DA task, the alignment difficulty may vary across different classes. Samples in the $\mathcal{D}_s$ that the classifier finds difficult to label can be referred to as difficult samples. For difficult samples, several of its categories may mostly be misclassified into other classes or sieved out because of low confidence in the classifier output, leading to severe category-balance bias in the selected samples and further negatively impacting DA. Methods for alleviating the class-balance problem for difficult samples are relatively novel and have achieved promising results.

Chen et al. [34] used the easy-to-hard transfer strategy (ETHS) to select reliable pseudo-label samples and then used adaptive prototype alignment (APA) to achieve cross-domain category alignment, as shown in Figure 5.

## Progressive Feature Alignment Network (PFAN)

**Figure 5.** The PFAN architecture. (Image: courtesy of Chen et al. [34]).

The network has the same construction as DANN [9] and consists of a feature extractor G, a label predictor F, and a domain discriminator D. In EHTS, by first computing the mean value of the features of each class in the $\mathcal{D}_s$ as the class prototype $c_k^S = \frac{1}{N_s^k} \sum_{(x_i^s, y_i^s) \in D_s^k} G(x_i^s)$, in which $c_k^S$ represents the class prototype of the k class of the $\mathcal{D}_s$, and then using the cosine similarity to estimate the distance between the $\mathcal{D}_t$ and each class of the $\mathcal{D}_s$ and finally setting a threshold value, the samples in the $\mathcal{D}_t$ that have phase degrees exceeding the threshold value are assigned pseudo-labels.

In the APA phase, the distance between each class in the $\mathcal{D}_s$ and $\mathcal{D}_t$ is defined as $d(c_k^S, c_k^T) = \|c_k^S - c_k^T\|^2$, and cross-domain class-level alignment is achieved by minimizing the APA loss. The APA loss and the total loss are defined as Equations (5) and (6), respectively:

$$L_{apa}(\theta_g) = \sum_{k=1}^{C} d\left(c_{k(I)}^S, c_{k(I)}^T\right) \tag{5}$$

$$\min_{\theta_g, \theta_f} \max_{\theta_d} \sum_{i=1}^{n_s} L_c\left(F\left(G\left(x_i^S; \theta_g\right); \theta_f\right), y_i^S\right) + \lambda L_d(\theta_g, \theta_d) + \gamma L_{apa}(\theta_g) \tag{6}$$

where $L_c$ is the standard cross-entropy loss, and $\lambda$ and $\gamma$ are the weights controlling the interaction between source categorization loss, domain confusion loss $L_d$, and APA loss. Zou et al. [29] mainly address the DA problem in semantic segmentation. To alleviate the issue of imbalanced classes caused by fixed thresholds, the paper sets a threshold $K_c$ for each class and gradually performs DA through self-step learning [35]. Zhang et al. [11] delved into the negative impact of inter-class imbalance of $\mathcal{D}_t$ samples on DA and proposed adaptive prediction calibration (APC) to mitigate the problem of hard classes by boosting hard classes, keeping common classes, and eliminating easy classes and introduced TE and SE (temporal fusion and self-fusion, respectively) to improve the reliability of prediction, as shown in Figure 6.

Recently, Liu et al. [36] suggested utilizing cyclic self-training as a replacement for standard self-training to tackle the issue of distribution bias in DA, as shown in Figure 7.

The network structure is the same as MCD [31] and contains a feature extractor and two classifiers. The difference is that the training alternates between two steps, the inner loop and the outer loop. In the inner loop, the $\mathcal{D}_t$ pseudo-labels are used to train the target classifier; in the outer loop, the shared representation is updated to boost the capability of the target classifier on the $\mathcal{D}_s$. To address the issue of noise amplification caused by high pseudo-label confidence, this study introduces an uncertainty metric derived from the information-theoretic Tsallis entropy. This metric can automatically minimize

the pseudo-label uncertainty without requiring any manual adjustment or setting of the confidence threshold.

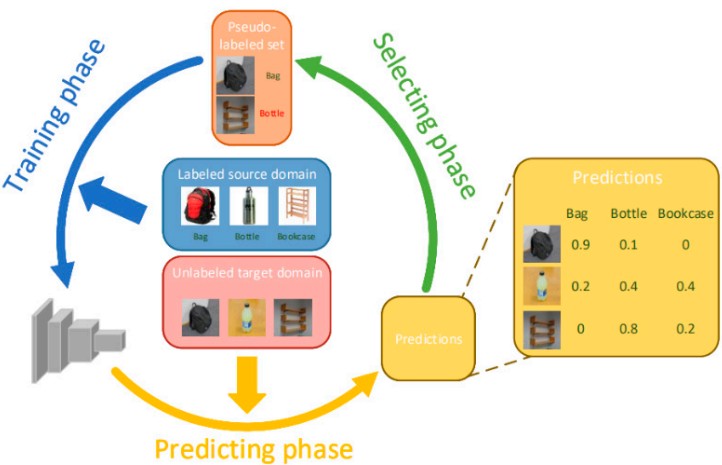

**Figure 6.** The hard class rectification for domain adaptation architecture. (Image: courtesy of Zhang et al. [11]).

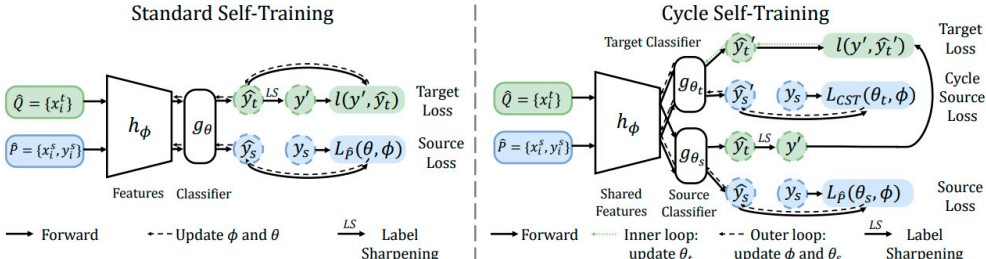

**Figure 7.** The cycle self-training for domain adaptation architecture. (Image: courtesy of Liu et al. [36]).

## 4. Application of Pseudo-Labeling in Domain Adaptation

Unlike pseudo-labeling generation methods, the application of pseudo-labeling in DA refers to the application of pseudo-labeling in traditional DA methods (e.g., adversarial-based, difference-based, and reconstruction-based methods, etc.). We classify them into two major categories: the application of pseudo-labeling in improving classifier discrimination and the application of pseudo-labeling in category feature alignment. The first category refers to methods that obtain classifiers with high generalization ability through supervised learning in the $\mathcal{D}_s$ and weakly supervised learning in the $\mathcal{D}_t$ that is labeled with high-quality pseudo-labels. The second category refers to methods that use pseudo-labeling to facilitate category feature alignment in the $\mathcal{D}_s$ and $\mathcal{D}_t$.

### 4.1. Application of Pseudo-Labeling in Improving Classifier Discrimination

Zhao et al. [37] integrated a DANN with a teacher–student network model [38] to learn feature representations with target differentiation using a consistency-forcing approach. It used prediction averaging and label sharpening to generate pseudo-labels for unlabeled $\mathcal{D}_t$ and introduced interpolation consistency into the unsupervised DA task to enhance the clarity of the decision boundaries. Zhang et al. [39] divided a CNN feature extractor into several blocks, each block being a set of CNN layers, as shown in Figure 8.

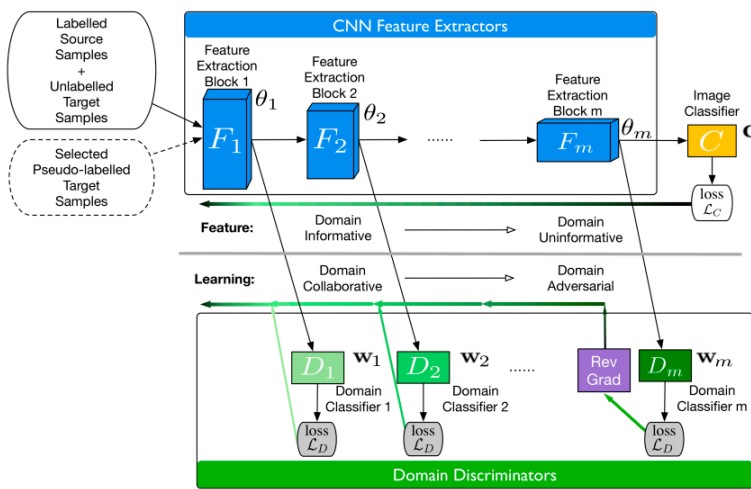

**Figure 8.** The CAN architecture. (Image: courtesy of Zhang et al. [39]).

The figure illustrates that each block for feature extraction includes a series of CNN layers. The domain classifier comprises several FC layers that serve to differentiate the domain to which each sample belongs. As the samples are propagated forward from the lower to the higher layers, the learned feature distribution changes smoothly from domain-relevant information to domain-independent information. Notably, the authors proposed an extension of the CAN method called incremental CAN (iCAN), i.e., incorporating the idea of self-training by leveraging the image classifier and the domain classifier from the previous training period, and iteratively choose a set of $\mathcal{D}_t$ samples with pseudo-labels. A dynamic thresholding method is employed to add these samples to the training set, achieving better results. The dynamic-thresholding-related settings are as follows in Equations (7)–(9).

$$T_C = \frac{1}{1 + e^{-\rho * A}} \tag{7}$$

$$A = \frac{1}{N_s} \sum_{i=1}^{N_s} I\left(y_i^s, \underset{c}{argmax}\ p_c(\mathbf{x}_i^s)\right) \tag{8}$$

$$I(a,b) = \begin{cases} 1, & if\ a = b \\ 0, & otherwise \end{cases} \tag{9}$$

where $T_C$ denotes the threshold value, in which $\rho$ was set to a fixed value of 3, $p_c(x_i^t)$ denotes the possibility of the i th sample, and $x_i^t$ pertains to class c. Xie et al. [40] put forward a method to evaluate the contribution of edge distributions (global) and conditional distributions (local) to the target task in the DA problem. Specifically, they learned the semantic representation of unlabeled $\mathcal{D}_t$ samples by aligning the labeled $\mathcal{D}_s$ examples and pseudo-labeled $\mathcal{D}_t$ examples. To alleviate the adverse effect of incorrect pseudo-labels, instead of aligning these newly acquired primes directly in each iteration, MSTN aligns exponentially moving average primes. Wang et al. [41] suggested a method called confidence-aware pseudo-labeling selection (CAPLS), which employs an iterative learning approach to gradually achieve domain alignment. Based on MMD, Kang et al. [42] introduced a difference measure called contrast domain difference (CDD) to explicitly model intra-class domain differences and inter-class domain differences. Recently, Chen et al. [43] further improved migration performance by using higher-order statistics for domain matching and using pseudo-labeled samples from the $\mathcal{D}_t$ to learn domain-invariant representations. Dong et al. [44] designed a confidence-anchor-induced pseudo-labels generator to mine the confidence pseudo-labels of the $\mathcal{D}_t$ by building confidence anchor groups and capturing consistent cross-domains by class-relationship-aware consistency loss inter-class relationships. Li et al. [45] defined an attention-aware transmission distance to measure domain differences using predictive feedback from an iterative learning classifier.

Yang et al. [46] proposed a bidirectional generation cross-domain generation framework by adding MMD loss and consistency loss to the loss function and pseudo-labeling the $\mathcal{D}_t$ data using the $\mathcal{D}_s$ classifier obtained from pretraining to implement a bidirectional cross-domain generation method. Hu et al. [47] preserved the category structure of the $\mathcal{D}_t$ by a duplex discriminator that also included classification tasks while aligning the overall features of the domain, as shown in Figure 9.

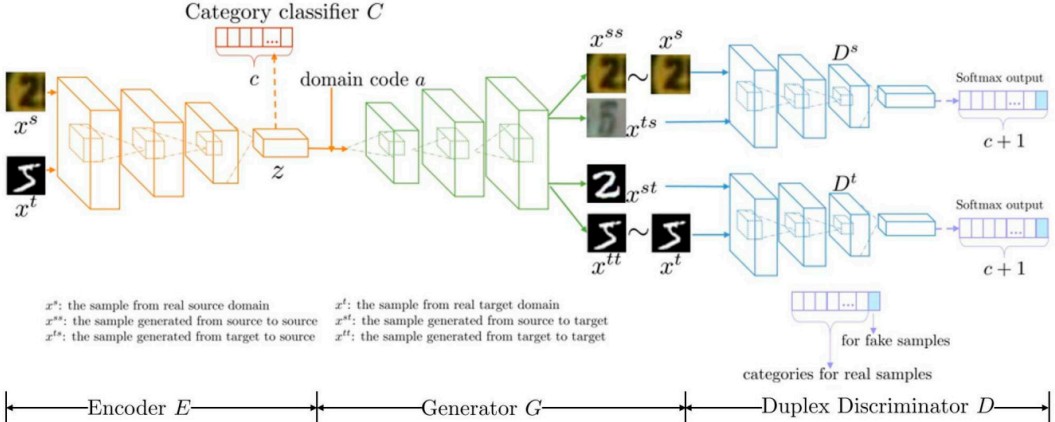

**Figure 9.** The DupGAN architecture. (Image: courtesy of Hu et al. [47]).

It consists of four parts: encoder $E$, generator $D$, duplex discriminators $D_s$ and $D_t$, and classifier $C$. The role of $E$ is to compress the image pixel-level features into $z$. Under the $D_s$ and $D_t$ constraint, the domain alignment is achieved by transforming the two-domain sample styles, while the $D_s$ and $D_t$ simultaneously classify the real image to preserve the class information of $z$. The $\mathcal{D}_t$ pseudo-labels missing during training are provided by Russo et al. [48], using GANS to introduce a symmetric mapping between the two domains and adding a class-consistency loss to enhance the structural stability and image quality of the reconstructed samples.

## 4.2. Application of Pseudo-Labeling in Category Feature Alignment

In recent years, the generative adversarial network (GAN) proposed by Goodfellow et al. [49] has been extensively employed in unsupervised learning. The GAN network primarily includes a generator and a discriminator. The generator in the GAN network generates synthetic samples using random noise, while the discriminator is responsible for distinguishing between real and synthetic samples. The GAN is trained by the strategy of maximum–minimum alternating optimization, and the ability of the discriminator to discriminate the authenticity is used as the "yardstick", thus generating samples that bear closer resemblance to the true samples. The GAN has been described in detail in [49–51], and interested researchers can refer to the above literature. The true and fake samples in the GAN can correspond to the $D_s$ and $\mathcal{D}_t$ in DA, respectively, and the generator corresponds to the feature extractor, while the discriminator is an implicit alignment scale. Due to the clear logic of GAN and its natural structural adaptation to the DA task, it has become a popular method in DA for learning transferable features that are domain invariant between the $D_s$ and $\mathcal{D}_t$.

The domain adversarial neural network (DANN) [9] is comprised of a feature extractor, a classifier, and a domain discriminator. It maximizes the domain confusion loss by using a gradient reversal layer (GRL) while minimizing the label prediction loss on the $D_s$ data to achieve feature alignment between the $D_s$ and $\mathcal{D}_t$. Unlike DANN, adversarial discriminative domain adaptation (ADDA) [10] uses separate feature extractors for each domain to capture more domain-specific information, aligns the $\mathcal{D}_t$ features toward the $D_s$ through a pretraining, fine-tuning training model, and finally tests the $\mathcal{D}_t$ samples using the $\mathcal{D}_t$ exclusive feature extractor and the $D_s$ classifier. The above two simple and

effective adversarial DA methods have become the basic architectures of many current DA methods [52–54]. Nonetheless, these two techniques only take into account the alignment of the marginal distribution between the $\mathcal{D}_s$ and $\mathcal{D}_t$ and do not consider the alignment of the conditional distribution (class-level alignment), so even if domain confusion is achieved, the classifier may perform poorly on the target task. As an analogy, by adversarial training, even with perfectly aligned marginal distributions, the feature space can still blend the characteristics of apples in the $D_s$ with those of oranges in the $\mathcal{D}_t$ [55].

To alleviate the above problems, more and more adversarial-based DA methods have started to consider category-level alignment, where the combination of adversarial training and pseudo-labeling methods is notable. Based on DANN, Zhang et al. [52] introduced center loss to achieve conditional distribution alignment. They proposed a method to deal with unlabeled samples in the $\mathcal{D}_t$. They used the predictions of the $D_s$ classifier to allocate pseudo-labels to each sample and defined the loss function as shown in Equation (10).

$$\min_{\theta_E} L_{ct} = \sum_{x_i \in \Phi(X_t)} \|E(x_i) - c_{\hat{y}_i}\|_2^2 \tag{10}$$

where $\hat{y}_i$ is the label of $x_i$ predicted by the classifier, and $c_{\hat{y}_i}$ denotes the center of the $i$ class. To alleviate the negative impact of incorrect pseudo-labeling during training, Zhang et al. [52] filtered a subset of $\mathcal{D}_t$ samples for training by means of a card-fixed threshold, and the filtering function is as follows in Equation (11).

$$\Phi(X_t) = \{x_i | x_i \in X_t \text{ and } max(p(x_i)) \geq T\} \tag{11}$$

where $p(x_i)$ is a K-dimensional vector, dimension $i$ corresponds to the predicted probability of class $i$, $max(p(x_i))$ is the possibility that sample $x_i$ pertains to the predicted class, and T is a fixed threshold. In [56], the selection of training samples is implicitly guided by pseudo-labels from the perspective of class-conditional domain alignment, focusing on the problem of intra-domain class imbalance and inter-domain class distribution shift.

Yu et al. [57] proposed transfer learning with a dynamic adversarial adaptation network (DAAN), which consists of three main components: labeled classifier, global domain discriminator, and local subdomain discriminator. The overall loss function is as follows in Equation (12).

$$L\left(\theta_f, \theta_y, \theta_d, \theta_d^c|_{c=1}^C\right) = L_y - \lambda\left((1 - \omega)L_g + \omega L_l\right) \tag{12}$$

where $\lambda$ is a constant value, while $\omega$ is a dynamic factor measuring the importance of $L_g$ and $L_l$. $L_y$, $L_g$, and $L_l$ denote classification loss, global loss, and local loss, respectively, where the pseudo-label of the $\mathcal{D}_t$ is also used in the calculation of $L_l$. Wang et al. [58] proposed an entropy-based adaptive reweighting adversarial DA method from the perspective of the conditional distribution, as shown in Figure 10.

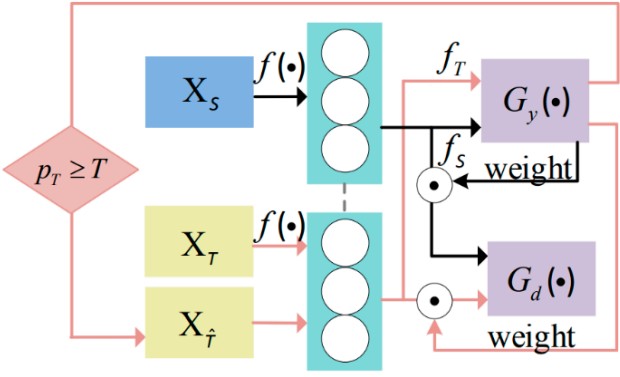

**Figure 10.** The self-adaptive reweighted adversarial DA architecture. (Image: courtesy of Wang et al. [58]).

To promote positive migration and curb negative migration, the method uses an entropy criterion to reveal the degree of sample transferability, which is then reweighted and fed back into the discriminative network to force the underlying distribution closer. In this paper, the loss function of domain adversarial training incorporates a conditional entropy term, and the weights assigned to different samples in the adversarial training are determined by the following Equations (13) and (14).

$$L_{adv}\left(\theta_f, \theta_d\right) = -\frac{1}{n_s + n_t} \sum_{x_i \in (D_s \cup D_t)} (1 + H_p)(L_d(G_d(f(x_i))), d_i) \tag{13}$$

$$\text{where} \quad H_p = -\frac{1}{C}\sum_{c=1}^{C} p_c log(p_c) \tag{14}$$

In addition to this, the authors use triplet loss to facilitate category-level alignment. Samples are randomly selected on the basis of the sampling approach. The pseudo-labels for the $\mathcal{D}_t$ are obtained by maximizing the posterior probability of the $D_s$ cross-entropy, which is gradually optimized as the model is trained. In addition, based on the intuitive consideration that images with high prediction scores are more likely to be correctly classified, only $\mathcal{D}_t$ samples with prediction scores above a certain threshold, T, are chosen for training in this paper, and the threshold is set as a constant in this paper. To avoid mislabeled target instances from propagating errors to the next iteration to disrupt the subspace learning process, Tanwani et al. [55] employ a network trained on $D_s$ data in the initial stages of training to predict pseudo-labels for unlabeled $\mathcal{D}_t$ and, then, retain only the most confident pseudo-labels for each category, resulting in a balanced mini-batch consisting of equal numbers of $D_s$ and $\mathcal{D}_t$ data for replacement sampling during training.

The construction of the graph convolutional adversarial network (GCAN) is proposed in Xinhong et al. [59]. The GCAN approach includes three alignment mechanisms: structure-aware, class-mass, and domain alignment. In class-mass alignment, class-mass alignment loss is computed using pseudo-labeled $\mathcal{D}_t$ features and labeled $D_s$ features to ensure that samples belonging to the same class from different domains are embedded closely. In order to develop the module for aligning the class centers of mass, the method uses a target classifier to assign pseudo-tags and obtains pseudo-tagged $\mathcal{D}_t$. Both labeled and pseudo-labeled samples are utilized to calculate the center of mass for each class. The DART (domain-against-residual transfer) network proposed by Fang et al. [60] comprises a deep feature extractor, a deep label classifier, and a domain classifier in its architecture. The entropy minimization method is used in computing the $\mathcal{D}_t$ label prediction loss by setting its loss function to Equation (15):

$$L_H = -\frac{1}{N_t}\sum_{i=1}^{N_t}\sum_{j=1}^{c} p(y_i^t = j|x_i^t)logp(y_i^t = j|x_i^t) \tag{15}$$

where $c$ represents the total number of classes, and $p(y_i^t = j|x_i^t)$ can be obtained by $p(y_i^t|x_i^t) = G_t\left(G_f(x_i^t)\right)$. Through minimizing the entropy penalty, the target classifier $G_t$ will self-adjust to expand the likelihood difference between predictions and predict more indicative labels accordingly. The alignment of the conditional distributions between the $D_s$ and $\mathcal{D}_t$ is accomplished in Cicek et al. [61] by incorporating an extra joint predictor. This predictor learns the distributions on the domain and class labels. The encoder is trained to deceive this predictor in the same class of samples for each domain. In [53], the confusion matrix is computed using a domain discriminator based on DANN as a way to correct the noise in the pseudo-labels. In [6], discriminator D discriminates the domain distribution along with the class distribution. Given that there exist some transferable regions between the $D_s$ and $\mathcal{D}_t$ images, we propose an attention module embedded in the GAN. In this way, we can remove as much background information as possible and further minimize the domain shift between the $D_s$ and $\mathcal{D}_t$. The corresponding experimental results can support our conclusion. To fully utilize the label information in the $\mathcal{D}_t$, we

present a straightforward yet effective approach to pseudo-label the unlabeled $\mathcal{D}_t$ samples. This idea can enhance the performance of classifier C while mitigating negative migration. Gu et al. [62] introduced an adversarial DA approach based on a spherical feature space and employed a Gaussian mixture model in the spherical space to obtain more robust pseudo-labels.

One of the most popular approaches in deep DA is to minimize the distributional discrepancy of domain features to achieve domain alignment, which employs deep neural networks to extract informative feature representations for the $D_s$ and $\mathcal{D}_t$ samples. Among them, two broad categories of domain distribution disparity metrics are commonly used: explicit and implicit. The explicit metrics are generally MMD distance, form-center distance, class-prototype distance, etc. In addition, implicit metrics are adversarial-based methods, popular learning, optimal transmission methods, etc. Fortunately, the pseudo-labeling approach can still be applied in a flexible manner in the aforementioned methods and lead to improved performance.

To achieve alignment of the conditional distributions of the $D_s$ and $\mathcal{D}_t$, Long et al. [63] modified the MMD to estimate the distance between the class-conditional distributions $Q_s(x_s|y_s = c)$ and $Q_t(x_t|y_t = c)$. The inter-class MMD distance is defined as follows in Equation (16).

$$\|\frac{1}{n_s{}^{(c)}} \sum_{x_i \in D_s{}^{(c)}} A^T x_i - \frac{1}{n_t{}^{(c)}} \sum_{x_j \in D_t{}^{(c)}} A^T x_j\|^2 \tag{16}$$

where the norm represents the $L_2$ norm, which is defined as the square root of the sum of the squared elements of a vector, A is the orthogonal transformation matrix, $D_s^{(c)}$ is the set of samples of class c in the $D_s$, and $n_s{}^{(c)} = \left| D_s^{(c)} \right|$. The same is true for the $\mathcal{D}_t$. Since the $D_t^{(c)}$ has no label, the authors incorporated the prediction of the $\mathcal{D}_t$ classifier directly as its pseudo-label in the computation.

Chadha et al. [54] enhanced the performance of ADDA by referring to the framework of semi-supervised GAN and exploiting the MMD loss. To fully leverage the discriminative information present in the distribution of labels, Luo et al. [64] put forward a method in which the features from the $D_s$ and $\mathcal{D}_t$ are mapped into a regenerated Hilbert kernel space, and the conditional distribution of the domains is represented by the conditional covariance operator in the kernel space. Then, the conditional kernel Bures (CKB) metric put forward in the paper is estimated and optimized based on the variance feedback. For each class, the semantic difference of that class between two domains is modeled using a multivariate Gaussian distribution that utilizes the inter-domain feature mean difference and the intra-class feature covariance on the $\mathcal{D}_t$, and then the $D_s$ features are augmented by randomly sampling semantic enhancement directions from the constructed distribution [65]. As a result of the absence of labels for the data in the $\mathcal{D}_t$, its pseudo-label is defined as $y'_{tj} = argmax_c P_{tj}^c$, where $P_{tj}^c$ is the softmax output of the $\mathcal{D}_t$ sample $x_{tj}$. Tanwisuth et al. [66] provides a framework for extracting class prototypes and aligning $\mathcal{D}_t$ features with them. Liang et al. [67], based on a nearest form-centered classifier, project the form centers of the $D_s$ and $\mathcal{D}_t$ features into an invariant subspace, where the pseudo-labels are computed using the feature transformation matrix and the maximum likelihood estimate of the $D_s$ expectation. Zhao et al. [68] defined a symmetric mirror loss based on Kullback–Leibler scatter to enhance the degree of domain alignment and followed an unsupervised discriminative clustering approach [69] to introduce auxiliary distributions as soft pseudo-labels. Li et al. [70] introduced a two-layer optimization strategy using pseudo-labels generated by the optimal classifier. With the purpose of boosting the accuracy of the pseudo-labels, Liang et al. [71] reduced the classifier bias by introducing auxiliary classifiers only for the $\mathcal{D}_t$ and incorporated the maximum prediction probability as a weight to the standard cross-entropy loss in Equations (17) and (18):

$$\hat{y}_i = \underset{k}{argmax}\ p_{i,k},\ i = 1, 2, \ldots, N_t \tag{17}$$

$$L_{pl}^{ours} = -\frac{\lambda}{N_{tu}} \sum_{i=1}^{N_{tu}} p_{i,\hat{y}_i} log p_{i,\hat{y}_i} \tag{18}$$

Sharma et al. [72] added a link to feature processing based on Gani et al. [9], as shown in Figure 11.

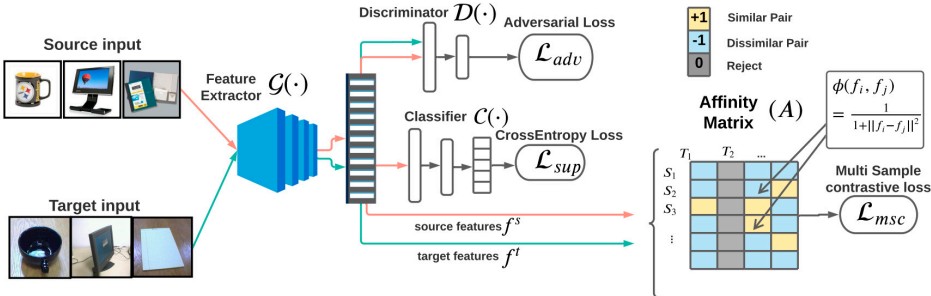

**Figure 11.** The ILA-DA architecture. (Image: courtesy of Sharma et al. [72]).

As in [9], the network includes an encoder, a classifier, and a discriminator. The encoder $G$ is shared between the $D_s$ and $\mathcal{D}_t$ and used to reduce the dimensionality of the data. After dimensionality reduction of the $D_s$ data by encoder $G$, the classifier C is used to generate a softmax predictive distribution of the categories, and then, supervised training is performed on the $D_s$ labeled data using standard cross-entropy loss. To achieve global feature alignment between the two domains, the authors train the domain discriminator $D$ using $L_D$ to classify the $D_s$ and $\mathcal{D}_t$ features and train $G$ using $L_{adv}$ to generate features for the confusion discriminator. In this way, domain-invariant features are extracted through a min-max training process between $L_D$ and $L_{adv}$.

The classification loss and the adversarial loss are defined as follows in Equations (19)–(21).

$$L_{sup} = E_{(x,y)\sim D^s} \left[ -log[C(G(x))]_y \right] \tag{19}$$

$$L_{adv} = E_{x\sim D^t}[-log D(G(x))] \tag{20}$$

$$L_D = -E_{x\sim D^s}[log D(G(x))] - E_{x\sim D^t}[log(1 - D(G(x)))] \tag{21}$$

Notably, The authors adopt the K-nearest neighbor (KNN) method to allocate pseudo-labels to the $\mathcal{D}_t$ samples ground on their similarity to nearby labeled $\mathcal{D}_s$ samples. To avoid some $\mathcal{D}_t$ samples being assigned incorrect pseudo-labels because some $\mathcal{D}_t$ samples may not have corresponding true $\mathcal{D}_s$ samples, the authors use a class-balanced small-batch sampling method to alleviate this problem. Finally, the correlation matrix is constructed based on similarity, and multiple-sample contrast loss is used to achieve class-level alignment of the $\mathcal{D}_s$ and $\mathcal{D}_t$ features.

Xu et al. [73] proposed a weighted optimal transfer strategy that uses spatial prototype information and intra-domain structure to reduce the negative transfer from samples near the decision boundary in the $\mathcal{D}_t$. Luo et al. [74] proposed a Riemannian manifold embedding and alignment framework that projects $D_s$ and $\mathcal{D}_t$ features into manifold space and uses a manifold metric to measure domain differences while taking into account both category-level alignment and global alignment.

In addition to the aforementioned mainstream methods, there are some DA methods using pseudo-labeling that have also achieved better results.

Hou et al. [75] proposed a source-free domain image translation (SFIT) method in which the model is split into two branches, one branch inputting $\mathcal{D}_t$ images and one branch using cycle-GAN as a generator to generate $D_s$-style images guided by the $D_s$ and $\mathcal{D}_t$ models.

The reconstruction-based approach data reconstruction refers to the addition of a data reconstruction task, typically using an autoencoder or generative adversarial network to ensure feature invariance during migration. Zhu et al. [76] proposed a new low-dimensional visual attribute (LDVA) coding method based on an autoencoder that can train end-to-end models for tasks such as DA, few-sample learning, and zero-sample learning.

In a data-enhancement-based approach, adversarial domain adaptation with domain mixup (DM-ADA), Xu et al. [77] proposes a mixup alignment. The features are gradually aligned by constructing some synthetic data, which serves as a bridge between the $D_s$ and $\mathcal{D}_t$. To realize domain alignment at the category level and ensure that features pertaining to the same category in both domains are mapped closely to the same latent space, the authors introduce classification loss to ensure category consistency between the decoded image and the input and mitigate the detrimental effects of mislabeling by filtering out samples with classification confidence below a certain threshold. Zhong et al. [78] introduced a general approach named E-MixNet, which improves the model performance by applying an enhanced mixup technique on labeled $D_s$ samples and pseudo-labeled $\mathcal{D}_t$ samples to curb the combinatorial risk in the target risk.

In a heterogeneous-based approach, Paolo et al. [79] propose a novel heterogeneous-distributed unsupervised DA method that focuses on the challenging setting of positive-unlabeled (PU) learning, where only positive and unlabeled examples are available. The method aims to enhance predictive models for a target domain by leveraging knowledge from a related source domain, even when the two domains are described with different feature spaces. The proposed method not only handles heterogeneous feature spaces but also efficiently distributes the workload to manage large volumes of data.

Existing unsupervised DA methods for time-series data have mainly centered on aligning the marginal distribution between the source and target domains. However, they tend to overlook the conditional distribution discrepancy, which can lead to misclassification in the target domain. He et al. [80] propose a novel method called ARADA-TK (attentive recurrent adversarial domain adaptation with top-k time-series pseudo-labeling) for unsupervised domain adaptation in time-series data. It focuses on learning domain-invariant representations by capturing temporal dependencies and reducing conditional distribution discrepancies.

## 5. Experience Evaluation

Given that image classification is a crucial task in various computer vision applications, the majority of the aforementioned algorithms were initially developed to address this issue. Therefore, in this section, we compare the current leading pseudo-labeling methods in unsupervised DA on the Office-31 classification dataset, showing how much benefit this method can bring to image classification.

The Office-31 dataset is widely used as a benchmark in visual DA, and it consists of 4652 images belonging to 31 object categories commonly found in office environments, such as laptops, filing cabinets, keyboards, etc. [81]. These images were primarily sourced from three different domains: Amazon (product images from online e-commerce websites), webcam (low-resolution images captured by webcams), and DSLR (high-resolution images captured by digital SLR cameras). There are 2817 images in the Amazon dataset, with an average of 90 images and one image background per category, 795 images in the webcam dataset, where the images show obvious noise, color, and white balance artifacts, and 498 images in the DSLR dataset. There are five objects in each category, and each object is pictured, on average, three times from different viewpoints.

Images are collected from online retailers (e.g., Amazon) and webcams (e.g., webcam) under various office-related categories. For the DSLR domain, images are taken using a high-quality DSLR camera. The collected images are resized and standardized to a fixed size, such as 224 × 224 pixels, to ensure uniformity and facilitate data processing. Then, they are divided into three domains: Amazon (A), webcam (W), and DSLR (D). Each image is associated with a domain label to indicate its source. To perform domain adaptation experiments, the dataset

is split into training and test sets. The training set contains images from the source domain (e.g., Amazon), along with their corresponding labels. The test sets are composed of images from the target domains (webcam and DSLR) without any labeled data.

Due to the variations in the parameters, experimental protocols, and tuning strategies used in different studies apart from pseudo-labeling, it is challenging to conduct a direct and fair comparison of all the methods. Therefore, we present comparison between the proposed method using pseudo-labels and an unsupervised DA method using only deep networks. In particular, we modularize these methods according to Sections 3 and 4 from the pseudo-label generation method and the application of pseudo-label adaptation in unsupervised domains to reflect the effectiveness of different modules. It is important to mention that all approaches are uniformly set to unsupervised DA of isomorphic closed sets with Resnet-50 as the framework, and we report the highest performance reported in the respective papers. The tables present the accuracy results of the unsupervised DA model (JAN) using only deep networks as a baseline, as shown in Tables 1 and 2.

**Table 1.** Classification Accuracy (%) Comparison for Different Pseudo-label Generation Methods on the Office-31 Dataset (ResNet-50).

| Generation Methods | Method ($\mathcal{D}_s \to \mathcal{D}_t$) | A → W | D → W | W → D | A → D | D → A | W → A | Avg |
|---|---|---|---|---|---|---|---|---|
| Baselines | JAN [82] | $85.4 \pm 0.4$ | $96.7 \pm 0.3$ | $99.7 \pm 0.1$ | $85.1 \pm 0.4$ | $69.2 \pm 0.4$ | $70.7 \pm 0.5$ | 84.6 |
| Single-classifier | SPL [17] | 92.7 | 98.7 | 99.8 | 93.0 | 76.4 | 76.8 | 89.6 |
| | CAT [18] | $94.4 \pm 0.1$ | $98.0 \pm 0.2$ | $100.0 \pm 0.0$ | $90.8 \pm 1.8$ | $72.2 \pm 0.6$ | $70.2 \pm 0.1$ | 87.6 |
| | PLUE-SFRDA [20] | 92.5 | 98.3 | 100.0 | 96.4 | 74.5 | 72.2 | 89.0 |
| Multi-classifier | SImpAI [27] | $97.9 \pm 0.2$ | $97.9 \pm 0.2$ | $99.4 \pm 0.2$ | $99.4 \pm 0.2$ | $71.2 \pm 0.4$ | $71.2 \pm 0.4$ | $89.5 \pm 0.3$ |
| | MCS [67] | 97.2 | 97.2 | 99.4 | 99.4 | 61.3 | 61.3 | 86.0 |
| | CAiDA [44] | 98.9 | 98.9 | 99.8 | 99.8 | 75.8 | 75.8 | 91.6 |
| Difficult samples | HCRPL [11] | $95.9 \pm 0.2$ | $98.7 \pm 0.1$ | $100.0 \pm 0.0$ | $94.3 \pm 0.2$ | $75.0 \pm 0.4$ | $75.4 \pm 0.4$ | 89.9 |

**Table 2.** Classification Accuracy (%) Comparison for Different Pseudo-label Application Scenario on the Office-31 Dataset (ResNet-50).

| Application Scenario | Method ($\mathcal{D}_s \to \mathcal{D}_t$) | A → W | D → W | W → D | A → D | D → A | W → A | Avg |
|---|---|---|---|---|---|---|---|---|
| Baselines | JAN [80] | $85.4 \pm 0.4$ | $96.7 \pm 0.3$ | $99.7 \pm 0.1$ | $85.1 \pm 0.4$ | $69.2 \pm 0.4$ | $70.7 \pm 0.5$ | 84.6 |
| Classifier discrimination | DIAL [52] | $91.7 \pm 0.4$ | $97.1 \pm 0.3$ | $99.8 \pm 0.0$ | $89.3 \pm 0.4$ | $71.7 \pm 0.7$ | $71.4 \pm 0.2$ | 86.8 |
| | MDD + Alignment [56] | $90.3 \pm 0.2$ | $98.7 \pm 0.1$ | $99.8 \pm 0.0$ | $92.1 \pm 0.5$ | $75.3 \pm 0.2$ | $74.9 \pm 0.3$ | 88.8 |
| | SRADA [58] | 95.2 | 98.6 | 100.0 | 91.7 | 74.5 | 73.7 | 89.0 |
| | DART [60] | $87.3 \pm 0.1$ | $98.4 \pm 0.1$ | $99.9 \pm 0.1$ | $91.6 \pm 0.1$ | $70.3 \pm 0.1$ | $69.7 \pm 0.1$ | 86.2 |
| | ALDA [53] | $95.6 \pm 0.5$ | $97.7 \pm 0.1$ | 100.0 | $94.0. \pm 0.4$ | $72.2 \pm 0.4$ | $72.5 \pm 0.2$ | 88.7 |
| | GAACN [6] | 90.2 | 98.4 | 100.0 | 90.4 | 67.4 | 67.7 | 85.6 |
| | RSDA-MSTN [62] | $96.1 \pm 0.2$ | $99.3 \pm 0.2$ | $100.0 \pm 0$ | $95.8 \pm 0.3$ | $77.4 \pm 0.8$ | $78.9 \pm 0.3$ | 91.1 |
| | TSA [65] | 94.8 | 99.1 | 100.0 | 92.6 | 74.9 | 74.4 | 89.3 |
| | PCT [66] | $94.6 \pm 0.5$ | $98.7 \pm 0.4$ | $99.9 \pm 0.1$ | $93.8 \pm 1.8$ | $77.2 \pm 0.5$ | $76.0 \pm 0.9$ | 90.0 |
| | MCS [67] | 97.2 | 97.2 | 99.4 | 99.4 | 61.3 | 61.3 | 86.0 |
| | Mirror [68] | $98.5 \pm 0.3$ | $99.3 \pm 0.1$ | $100.0 \pm 0.0$ | $96.2 \pm 0.1$ | $77.0 \pm 0.1$ | $78.9 \pm 0.1$ | 91.7 |
| | i-CDD [70] | $95.4 \pm 0.4$ | $98.5 \pm 0.2$ | $100.0 \pm 0.0$ | $96.3 \pm 0.3$ | $77.2 \pm 0.3$ | $78.3 \pm 0.2$ | 90.9 |
| | ATDOC [71] | 94.6 | 98.1 | 99.7 | 95.4 | 77.5 | 77.0 | 86.1 |
| | ILA-DA [72] | 95.7 | 99.2 | 100.0 | 93.3 | 72.1 | 75.4 | 89.3 |
| | RWOT [73] | $95.1 \pm 0.2$ | $94.5 \pm 0.2$ | $99.5 \pm 0.2$ | $100.0 \pm 0.0$ | $77.5 \pm 0.1$ | $77.9 \pm 0.3$ | 90.8 |
| | Fine-tuning [75] | 91.8 | 98.7 | 99.9 | 89.9 | 73.9 | 72.0 | 87.7 |
| | E-MixNet [78] | $93.0 \pm 0.3$ | $99.0 \pm 0.1$ | $100.0 \pm 0.0$ | $95.6 \pm 0.2$ | $78.9 \pm 0.5$ | $74.7 \pm 0.7$ | 90.2 |

**Table 2.** *Cont.*

| Application Scenario | Method ($\mathcal{D}_s \rightarrow \mathcal{D}_t$) | A → W | D → W | W → D | A → D | D → A | W → A | Avg |
|---|---|---|---|---|---|---|---|---|
| | iCAN [39] | 92.5 | 98.8 | 100.0 | 90.1 | 72.1 | 69.9 | 87.2 |
| | CAPLS [41] | 90.6 | 98.6 | 99.6 | 88.6 | 75.4 | 76.3 | 88.2 |
| | CAN [42] | 94.5 ± 0.3 | 99.1 ± 0.2 | 99.8 ± 0.2 | 95.0 ± 0.3 | 78.0 ± 0.3 | 77.0 ± 0.3 | 90.6 |
| Category feature alignment | HoMM [43] | 91.7 ± 0.3 | 98.8 ± 0.0 | 100.0 ± 0.0 | 89.1 ± 0.3 | 71.2 ± 0.2 | 70.6 ± 0.3 | 86.9 |
| | CAiDA [44] | 98.9 | 98.9 | 99.8 | 99.8 | 75.8 | 75.8 | 91.6 |
| | ETD [45] | 92.1 | 100.0 | 100.0 | 88.0 | 71.0 | 67.8 | 86.2 |
| | BDG [46] | 93.6 ± 0.4 | 99.0 ± 0.1 | 100.0 ± 0. | 93.6 ± 0.3 | 73.2 ± 0.2 | 72.0 ± 0.1 | 88.5 |

## 6. Challenges and Future Directions

Although pseudo-labeling methods have a large number of applications in deep DA with good results, there are still some problems, and we present them and indicate potential areas for future research.

(1) There is a lack of a common, universal indicator to evaluate the quality of pseudo-labels.

In DA with pseudo-labeling methods, the quality of pseudo-labels can directly affect the effectiveness of DA. Through literature research, we found that only a few researchers have analyzed the quality of pseudo-labels: Zhang et al. [11] portrayed the variation of pseudo-label accuracy with training cycles in the ablation experiment section, while Liu et al. [36] plotted ROC curves and used the AUC metric to quantify pseudo-label effectiveness. We believe that a generalized metric for evaluating the quality of pseudo-labels would be useful for the development of DA methods that use pseudo-labels.

(2) Cross-domain issues affect the quality of pseudo-labels.

Unlike semi-supervised learning where the training and test sets obey the same distribution, DA faces the challenge of cross-domain distribution shift. Liu et al. [36] experimentally confirmed that in the cross-domain case (where the pseudo labels of the $\mathcal{D}_t$ are generated by the $D_s$ model), the quality of the obtained pseudo-labels is lower than when the $D_s$ and $\mathcal{D}_t$ obey the same distribution. Moreover, the difficulty (inter-domain distance) of different migration tasks is different, and finding a pseudo-labeling method that is applicable to different difficulty DA problems is a worthwhile research direction.

(3) The dataset is more homogeneous, while the real scenario is more complex.

At present, the public datasets used for DA are generally Digits, Office-31, VisDA-2017, etc. It is fair to compare with public datasets for theoretical studies, which is beneficial to the theoretical development in this direction. However, it is possible that the proposed method performs better only on the mentioned datasets. Further expansion of more complex public datasets in the future will facilitate the application of DA methods in real scenarios.

(4) Research has mainly focused on classification problems, and there is a lack of research on other DA problems.

The current application of the pseudo-labeling method for DA mainly solves the classification problem, and its application in semantic segmentation DA, weakly supervised DA, and domain generalization can be further tried in the future.

## 7. Conclusions

Deep DA is a research area with important real-world applications. The successful application of pseudo-labeling methods in deep DA has further contributed to its rapid development. This review classifies DA methods using pseudo-labeling into self-training-based methods, divergence-based methods, adversarial-based methods, difference-based methods, and other methods. Finally, we discuss the challenges faced by pseudo-labeling in DA applications and some directions that deserve further research in the future.

**Author Contributions:** Conceptualization, Y.L. and L.G.; methodology, Y.L.; software, Y.G.; validation, Y.L. and L.G.; formal analysis, Y.L.; investigation, Y.G.; resources, Y.L.; data curation, L.G.; writing—original draft preparation, Y.L.; writing—review and editing, L.G.; visualization, Y.G.; supervision, Y.G.; project administration, Y.L.; funding acquisition, Y.L. All authors have read and agreed to the published version of the manuscript.

**Funding:** This research was funded by [the National Natural Science Foundation of China] grant number [62071006].

**Data Availability Statement:** Data available in a publicly accessible repository that does not issue DOIs. Publicly available datasets were analyzed in this study. This data can be found here: [https://paperswithcode.com/dataset/office-31].

**Conflicts of Interest:** The authors declare no conflict of interest.

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
