# Peer review of "Pseudo Labels for Unsupervised Domain Adaptation: A Review"

_electronics, doi:10.3390/electronics12153325_

Round 1

Reviewer 1 Report

(1) In this paper, data mining methods based on pseudo-labeling are divided into self-training methods, divergent methods, adversarial methods, and differentiation-based methods, etc. It is suggested that the application of pseudo-label generation methods and pseudo-label adaptive methods in unsupervised data mining can be further explored, so as to find more suitable methods to optimize model performance.

(2) Considering that the real scenario is more complex, it is suggested that the author should use the same parameters, experimental schemes and adjustment strategies as much as possible to ensure the fairness and accuracy of the comparison results.

(3) This paper summarizes the pseudo-label generation methods based on single classifiers and multiple classifiers, and the measures taken to deal with the problem of sample imbalance. Are there other ways to deal with sample imbalance?

(4) How effective is the application of pseudo-annotation in classification feature alignment and improved feature recognition? Can you also try to apply the pseudo-annotation method to other unsupervised data mining tasks? Please explain in the text.

Reviewer 2 Report

In this paper, the authors proposed a survey for domain adaptation regarding the pseudo labelling of unlabelled instances. In the literature domain adaptation surveys exist but the focus here is to provide an overview of a more specific problem afflicting domain adaptation problems. 

The authors could better discuss also the following latest pseudo-labeling methods in transfer learning settings:

- DOI: 10.1109/BigData55660.2022.10020270

- DOI: 10.1007/s10489-022-04176-x

- DOI: 10.1002/int.22930

Formatting issues:

- acronyms and citations should be presented with 1 white space while currently they are connected to the text.

For instance:

"Domain Adaptation(DA)" ==> "Domain Adaptation (DA)"

"Tan et al.[1]provided" ==> "Tan et al. [1] provided"

...

- Figures (for instance Fig. 1 and 2) currently are poor in quality. Consider improving (for instance by exporting the image as vectorial PDF)

Reviewer 3 Report

This paper is well-organized, but I have a few concerns that I hope the authors could address carefully:

1. The novelty of this work appears to be incremental. Could you kindly clarify the major contributions to the field?

2. This work only uses accuracy as the metric. It would be beneficial to involve different metrics to compare the performance.

3. It would be necessary to include average values along with their associated standard deviations in the tables to facilitate comparison.

4. The data preprocessing steps are not clearly described. Could you provide more details on how the data was prepared for analysis?

5. The dataset used in the paper seems relatively small. Is it possible to validate your approach using a larger scale dataset?

6. The definition of inter-class MMD distance in this paper is unclear. Could you clarify what norm Eq 16 represents?

I look forward to your responses and the subsequent revisions of the paper.

The writing English needs to be improved. 
